# Smart Insulin Pen in Pregnant Women with Type 1 Diabetes: An Encouraging Case Series

**DOI:** 10.3390/healthcare13010038

**Published:** 2024-12-29

**Authors:** Veronica Resi, Alessia Gaglio, Yana Pigotskaya, Amelia Caretto, Emanuela Orsi, Valeria Grancini

**Affiliations:** 1Endocrinology Unit, Fondazione IRCCS Ca’ Granda Ospedale Maggiore Policlinico, 20122 Milan, Italy; alessia.gaglio@policlinico.mi.it (A.G.); yana.pigotskaya@unimi.it (Y.P.); emanuela.orsi@policlinico.mi.it (E.O.); valeria.grancini@policlinico.mi.it (V.G.); 2Diabetes Research Institute, IRCCS Ospedale San Raffaele, 20132 Milan, Italy; caretto.amelia@hsr.it

**Keywords:** type 1 diabetes, pregnancy, smart insulin pen, CGM

## Abstract

**Background:** The management of type 1 diabetes in pregnancy with new technologies is challenging. Sometimes the complexity of new-generation systems such as “continuous subcutaneous insulin infusion, CSII” and patient or provider preference do not allow their use, so women with type 1 diabetes in pregnancy continue to be treated with subcutaneous multiple-injection insulin therapy using pens. Smart insulin pens are new tools that allow for data collection on insulin dose and time of administration and have additional connectivity features. **Objective:** To retrospectively describe the use of a smart insulin pen coupled with rt-CGM (InPen^TM ^system) in three pregnancies complicated by type 1 diabetes. **Methods:** Participants used the InPen^TM ^system in pregnancy and consented to analysis of glycaemic data and pregnancy outcome. **Results:** An increase in pregnancy specific time-in-range glucose was observed in the three patients related to the duration of insulin action, insulin sensitivity factors, and a pre-set target glucose range for pregnancy. No diabetic ketoacidosis or severe hypoglycaemia occurred. **Conclusions:** We describe practical considerations in three pregnant patients with type 1 diabetes where the InPen^TM ^system was used with suggestive improvements in the time-in-range.

## 1. Background

Pregnancy complicated by type 1 diabetes has always been a therapeutic and management challenge, with neonatal and maternal outcomes still not comparable to those in the non-diabetic population. Glycaemic control is the most important variable associated with good or poor pregnancy outcomes [1]. As a result, the American Diabetes Association (ADA) recommends strict glycaemic targets for pregnant women with type 1 diabetes including an A1C target of <6.5% without significant hypoglycaemia and self-monitoring of both fasting and postprandial blood glucose (SMBG) using an insulin pump or basal bolus insulin [2]. To date, the literature on smart insulin pens has focused on assessing patient preference, ease of use, and technical accuracy, investigating the preference choice for people with diabetes [3]. Recent studies have shown improved insulin adherence through a reduction in the number of missed bolus doses, better mealtime dosing, and increased TIR in people with type 1 diabetes using smart insulin pens with connectivity in a real-world setting [4]. Munshi and colleagues suggested that use of a smart pen with connectivity may reduce the gap between patient-reported and actual adherence [5]. The use of technology in the management of diabetes is now common practice, and the use of new tools in pregnancy is becoming increasingly necessary. There is strong evidence from the “Continuous glucose monitoring in pregnant women with type 1 diabetes“ (CONCEPT study) that multiple daily injections (MDIs) plus continuous glucose monitoring (CGM) users compared to pump plus CGM users were more likely to achieve targets for A1c and have an increased time-in-range (TIR) by 24 weeks gestation [6]. The international consensus for CGM-derived glycaemic targets states that pregnant women with type 1 diabetes should have >70% ps-TIR (63–140 mg/dL), <25% of time-above-range (TAR > 140 mg/dL), and 5% of time-below-range (TBR <4% of time <63 mg/dL and <1% of time <54 mg/dL) [7]. The strict glycaemic targets required in pregnancy are difficult to achieve [8]. The InPen^TM^ is claimed to be the first FDA-cleared and CE-marked smart insulin pen that integrates with real-time CGM via a convenient smartphone app [9]. The use of InPen^TM^ paired with CGM (Guardian G4-InPen^TM^ system) during pregnancy has not been reported previously. The aim of these case series was to describe the use of smart insulin pens with connectivity for diabetes management in pregnancy.

## 2. Research Design and Methods

This is a retrospective case series conducted on three pregnant women with type 1 diabetes, all of whom received the same intervention, with no control group, so it cannot be compared as a case series. We have retrospectively collected the demographics, medical characteristics, and pregnancy outcomes of three women with type 1 diabetes who were managed with an InPen^TM^ system during pregnancy. The system includes a reusable pen compatible with insulin lispro and insulin as part of rapid-acting insulin cartridges [10]. After the insulin cartridge is installed, the device sends real-time data via Bluetooth to an InPen^TM^ application (app). The system was installed by an expert physician on each patient’s smartphone, and the smart insulin pen and cell phones were paired. Patients were instructed on how to use the cell phones and smart insulin pen. The InPen ^TM^ has additional features such as a bolus dose calculator, detection of prime dose versus actual dose, reminder alerts, active insulin on board (IOB) monitoring, and integration with CGM [11]. All patients were managed by a team of physicians, dietitians, and gynaecologists with expertise in type 1 diabetes management. As part of our standard care in the management of type 1 diabetes in pregnancy, target blood glucose ranges and active insulin time (AIT) settings were adjusted according to recommendations, individual preferences, and our clinical experience. Settings were chosen to achieve the goals of pregnancy-specific targets. The scheduled antenatal visits for all three people were twice a month during the pregnancy time at a local outpatient clinic. This study was approved by the institutional ethics committee (Comitato Etico Territoriale Lombardia 3 -ID 5378 –5378_16.10.2024_P) and complied with the Declaration of Helsinki and good clinical practice guidelines. All participants provided written informed consent. HbA1c, continuous glucose monitoring data with recommended pregnancy-specified target ranges, the insulin–carbohydrate ratios, and insulin sensitivity factors during each trimester of pregnancy are summarised in Table 1.

### 2.1. Case 1

This is the second pregnancy of a 30-year-old woman with type 1 diabetes since the age of 6. She had non-proliferative diabetic retinopathy since 2011. In 2018, the patient had tried continuous subcutaneous insulin infusion (CSII) therapy which was discontinued due to discomfort. The first pregnancy was in 2019 and resulted in a dystocic birth at 37 weeks gestation, a 3975 g male, due to altered cardiotocographic tracing. During the first pregnancy, the patient was on controlled therapy with MDI and intermittently scanned continuous glucose monitoring (isCGM), and the pregnancy described in this report was unplanned. Prior to conception, she had an A1C of 6.7% and a pre-pregnancy weight of 70 kg (BMI 23.4 kg/m^2^). She was treated with MDI insulin with an ins/cho ratio (ICR) of 1/8 at breakfast, 1/7.5 at lunch, and 1/10 at dinner before each meal plus insulin glargine at a dose of 9 units nightly at bedtime. She was still monitoring her glucose levels with both SMBG and isCGM. In the month before conception, her overall mean sensor glucose was 134 mg/dL with a coefficient of variation (CV) of 41%, indicating unhealthy glycaemic control. The patient was referred to our Centre at 13 weeks’ gestation. In the first trimester of pregnancy, her isCGM readings were as follows: ps-TIR 68% of the time, ps-TBR < 63 mg/dL 6% of the time and <54 mg/dL 2% of the time, and ps-TAR 24% of the time. At 15 weeks’ gestation, therapy was switched from MDI and isCGM to the InPen^TM^ integrated with the Guardian 4 CGM system. The pregnancy-specified target and duration of insulin action (IOB) were established (2.5 h). The ICR and the insulin sensitivity factors used to calculate dose recommendations were also specified in the system. PS-TIR increased to 74% during the first 2 weeks of InPen^TM^ use. She noticed an increase in her insulin requirements, with postprandial hyperglycaemia (both morning and afternoon), and the ICR was reduced to 6.5 g in the morning and 5.5 g in the afternoon for more bolus insulin. At 30 weeks’ gestation she achieved ps-TIR over 89%, ps-TAR 7%, ps-TBR 4%, and GMI 5.9% without severe hypoglycaemia. The time-above-range of 19% observed in the second trimester was decreased to 7% in the third trimester. The total daily insulin dose was adjusted during the pregnancy. Good control with time-below-range <5% was maintained in the third trimester (Figure 1). At the end of pregnancy, her gestational weight gain was 14 kg. The patient gave birth to a male infant at 38 weeks’ gestation with a weight of 3980 g and a length of 53 cm (Apgar at 1′ e 5′, 9, and 10, respectively). His size was large for gestational age (LGA) with no congenital malformations; the infant required temporary monitoring for hyperbilirubinemia 3 days after birth which resolved with phototherapy. At 4 weeks postpartum, the infant was healthy, and the mother had maintained stable glycaemic control (mean glucose level 115 ± 49 mg/dL, CV 43.0%, and GMI 6.1%).

### 2.2. Case 2

A 36-year-old woman, presented at 8 weeks’ gestation in January 2024. She had type 1 diabetes since 2003 and Hashimoto’s thyroiditis since 2010. She was treated with MDI and an intermittently scanned glucose meter (isCGM). She did not plan the pregnancy, she was on MDI insulin with 2 units at breakfast, 3 units at lunch, 4 units at dinner, and insulin glargine U300 10 units at bedtime. She has no knowledge of CHO counting. Her pre-pregnancy weight was 60 kg, her pre-pregnancy BMI was 21.5 kg/m^2^, and her HbA1c a month prior to conception was 6.8%. In the first trimester of pregnancy, her isCGM readings were as follows: ps-TIR 64% of the time, ps-TBR < 63 mg/dL 10% of the time and < 54 mg/dL 4% of the time, and ps-TAR 22% of the time. At 10 weeks’ gestation, therapy was upgraded with the InPen^TM^ integrated with the Guardian 4 CGM system. The pregnancy-specified target and duration of insulin action (IOB) were set up (2.5 h). She was not able to apply the CHO counting method so the dose calculator of the system mobile app was set with fixed doses. The dose calculator of the InPen ^TM^ system helped the patient in adjusting insulin doses at meals and corrected hyperglycaemia, avoiding hypoglycaemia due to excessive correction boluses. At 28 weeks’ gestation she achieved ps-TIR over 84%, ps-TAR 12%, ps-TBR 4%, and GMI 5.6% with a reduction in hypoglycaemic events. The total daily insulin dose was adjusted during the pregnancy. At 37 weeks’, her HbA1c was 5.4% ps-TIR 90%, ps-TAR 6%, and ps-TBR 2 + 2% with an overall mean sensor glucose of 97 mg/dL and CV of 24.5% (Figure 2). The total daily dose was 29.7 units (15.6 units of a total rapid insulin dose and 14.1 units of insulin glargine). At the end of pregnancy, her gestational weight gain was 10.5 kg. The patient gave birth to a female infant at 39 + 1 weeks’ gestation with a weight of 3480 g and a length of 51 cm, appropriated for gestational age (AGA) with no perinatal complications.

### 2.3. Case 3

A 34-year-old female patient, followed up in another centre, with type 1 diabetes since the age of 14, with a low level of knowledge about the disease. She had been addressed at our facility at 8 weeks gestation in an unplanned pregnancy. She was treated with MDI and an isCGM sensor and has always refused insulin pump therapy, with pre-pregnancy poor metabolic compensation (HbA1c 7.3%) with a weight of 55.5 kg and a pre-pregnancy BMI of 18.7 kg/m^2^. The patient was treated with rapid-acting insulin analogue glulisine, which was immediately switched to insulin lispro. It was recommended to switch to the InPen^TM^ to simplify the management of insulin therapy and to integrate glucose monitoring data to make it easier for the patient to manage her therapy. The total daily dose was 45.8 units distributed as 3.5 at breakfast, 1.5 at snack, 8 units at lunch, and 8 units at dinner, with 22 units of insulin glargine U100. The pregnancy-specified target and duration of insulin action (IOB) were set up (2.0 h). She was not able to apply the CHO counting method, so the dose calculator of the system mobile app was set with fixed doses. Due to low compliance, at the second trimester, the time of sensor use was not optimal and ps-TIR was not sufficiently low: ps-TIR 68% and ps-TAR 28%; nonetheless, the use of CGM helped in reducing ps-TBR 2 + 2%. The dose calculator of the InPen^TM^ system helped the patient in adjusting insulin doses at meals and correct hyperglycaemia, avoiding hypoglycaemia due to excessive correction of boluses. Sensor glucose profiles helped the patient to promptly identify post-prandial hyperglycaemia dose reminders using the InPen^TM^ app. Pregnancy-specific TIR increased from 68% in the second trimester to 76% in third trimester, and CV reduced from 36.3% in first trimester to 30.4% in third trimester (Figure 3). At the third trimester, we determined an increasing need for rapid-acting insulin, due to insulin resistance from the pregnancy. The patient attended regular consultations with our dietician to ensure adequate CHO consumption and to limit foods with a high glycaemic index. At the end of the pregnancy, the weight gain was 12 kg since the beginning. At 37 weeks of pregnancy, the patient was admitted for polyhydramnios (amniotic fluid index 26); a caesarean section was performed, and the patient gave birth to a male infant with a weight of 3680 g and a length of 52 cm; his size was large for his gestational age (LGA). At 4 weeks postpartum, the mother had a stable glycaemic control.

## 3. Discussion

Efficacy comparisons between new-generation systems of technologies versus traditional systems are beginning to provide some interesting data even in pregnancy complicated by type 1 diabetes. Sometimes perceived complexity and patient/provider preference do not allow for the use of infuse systems such as CSII or the Hybrid Closed Loop System “HCL”. The ADA includes connected insulin pens in the Medical Standards of Care recommendations for insulin therapy as a solution to potentially help patients on injection therapy with dose recall, dose recommendation, and dose titration [2]. The ADA notes that there is no “one-size-fits-all” approach to technology use in diabetes care, and individuals should be supported in technology options that best match their circumstances, desires, and needs [2]. Therefore, smart insulin pens may have the potential to improve adherence and achieve glycaemic targets in type 1 diabetic pregnancies, which are important for maternal and foetal outcomes. This system uses user-defined settings: personalised insulin-to-carbohydrate ratios, including duration of insulin action, insulin sensitivity factors, and a pre-set target glucose range for pregnancy. The ability to process these operations and the duration of insulin action (IOB) is an extremely useful feature. In addition, the system offers reminders to check blood glucose, for missed doses, and for long-acting insulin dose reminders. Integrated rt-CGM data can provide a useful overview for healthcare professionals and patients using this system. The InPen ^TM^ system during unplanned pregnancy with adverse obstetrical or neonatal outcomes can help in optimising glucose control and reducing hypoglycaemia. The bolus calculator of the InPen ^TM^ system can quickly adapt to the constant changes in rapid-acting insulin that occur during pregnancy. For the three cases, we have set the system to the pregnancy-specific target according to the ATTD consensus CGM target recommendations, which suggest a TIR for 63–140 mg/dL of >70% (16 h, 48 min), a TAR for >140 mg/dL of <25% (6 h), a TBR for <63 mg/dL of <4% (1 h), and a TBR for <54 mmol/L of <1% (15 min) [7]. The insulin sensitivity factor was calculated for each patient for the three trimesters as shown in Table 1. Although carbohydrate counting is recommended for people with type 1 diabetes by the American Diabetes Association [2], two out of the three cases presented did not use CHO counting. This may be due to lack of education or non-compliance. Carbohydrate counting is a marker of better adherence in general; in this observation, using the tools provided by the system, even in women who did not use CHO counting, the patients were able to manage the patterns of low and high blood glucose and adjust their insulin needs throughout their pregnancy without severe hypoglycaemia or ketoacidosis. As stated by Bellido and colleagues it is important to incorporate the concept of ps-TIR into educational diabetes programs to increase the engagement of women in achieving pregnancy-related glycaemic goals [12]. Our patients achieved the recommended 70% time-in-range between the second and the third trimester of pregnancy with the system under use, which is still too late for optimal neonatal outcomes. We did not use validated questionnaires to assess patient satisfaction, as this was a retrospective analysis of three cases. We will be interested in profiling patients with diabetes in pregnancy who may benefit from using this system.

## 4. Conclusions

In conclusion, in our case series, with the use of this new technological tool in women with type 1 diabetes in pregnancy, we observed an improvement in ps-TIR in the late second trimester of pregnancy. In particular, the use of this technology could find a role in planning a pregnancy with type 1 diabetes as an educational and interventional tool in women with type 1 diabetes on MDI, as the system’s features provide useful tools for optimising insulin therapy to achieve the ambitious target metrics in the pre-pregnancy and early pregnancy state. To characterise the benefits of this system in such a complex population, greater real-world data are needed.

## Figures and Tables

**Figure 1 healthcare-13-00038-f001:**
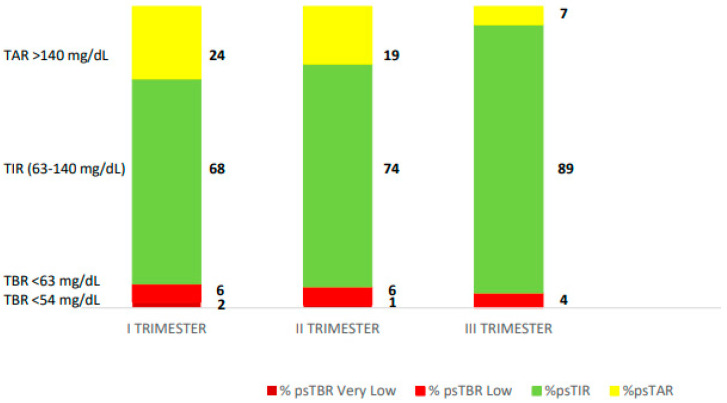
The diagram displays the average pregnancy-specified time-in-range (%ps-TIR), average pregnancy-specified time-above-range (% ps-TAR), and average pregnancy-specified time-below-range (% ps-TBR) in each trimester for RT-CGM.

**Figure 2 healthcare-13-00038-f002:**
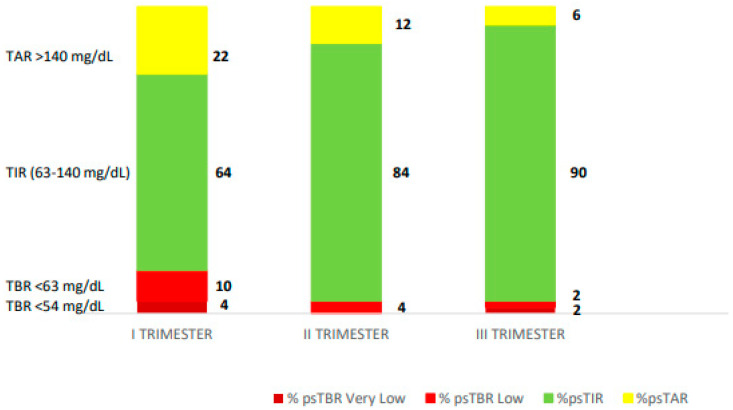
The diagram displays the average pregnancy-specified time-in-range (%ps-TIR), average pregnancy-specified time-above-range (% ps-TAR), and average pregnancy-specified time-below-range (% ps-TBR) in each trimester for RT-CGM.

**Figure 3 healthcare-13-00038-f003:**
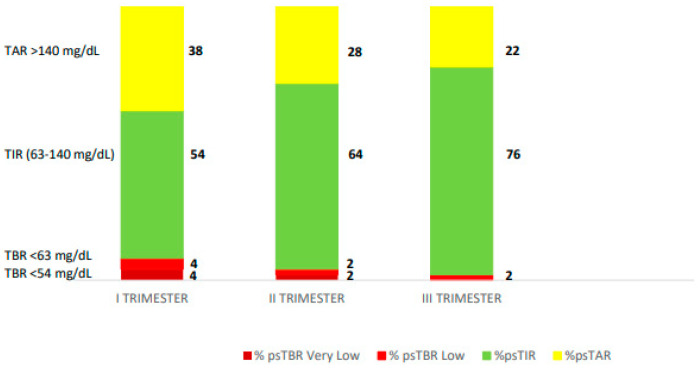
The diagram displays average the pregnancy-specified time-in-range (%ps-TIR), average pregnancy-specified time-above-range (% ps-TAR), and average pregnancy-specified time-below-range (% ps-TBR) in each trimester for RT-CGM.

**Table 1 healthcare-13-00038-t001:** Selected baseline clinical characteristics, glycaemic values, and insulin requirement during pregnancy for each trimester.

	Case 1	Case 2	Case 3
Age (years)	30	36	34
Diabetes duration (years)	24	21	20
No. of previous pregnancies (n)	1	0	0
Microvascular complications	1	0	0
Planned pregnancy (Y/N)	N	N	N
Pre-pregnancy BMI (kg/m^2^)	23.4	21.5	18.7
Pre-pregnancy HbA1c (%)	6.7	6.8	7.3
**I trimester (weeks 8–12)**			
Mean glucose level (mg/dl)	116 ± 36	125 ± 24	132 ± 16
HbA1c (%)	6.7	6.8	7.3
Coefficient of variation (CV %)	41.0	34.0	36.3
Total daily bolus insulin, mean (IU/day)	20.8	9.7	21.4
Total daily basal insulin, mean (IU/day)	9.4	10.8	22.6
Total daily insulin—IU/kg/day	0.4 ± 0.1	0.31± 0.15	0.7 ± 0.1
Insulin sensitivity factor (IU/mg/dL)	1:55	1:85	1: 40
Sensor Use (%)	89	90	78
**II trimester (weeks 13–26)**			
Mean glucose level (mg/dL)	108 ± 42	89 ± 13	119 ± 28
HbA1c (%)	6.4	5.6	6.4
Coefficient of variation (CV %)	39.5	29.2	32.3
Total daily bolus insulin, mean (IU/day)	27.8	12.5	27.8
Total daily basal insulin, mean (IU/day)	21.6	14	21.6
Total daily insulin—IU/kg/day	0.6 ± 0.2	0.38 ± 0.1	0.82 ± 0.1
Insulin sensitivity factor (IU/mg/dL)	1:36	1:75	1:36
Sensor Use (%)	98	96	82
**III trimester** **(weeks 27–40)**			
Mean glucose level (mg/dL)	104 ± 33	97± 11	102 ± 14
HbA1c (%)	5.9	5.4	6.0
Coefficient of variation (CV %)	30.2	24.5	30.4
Total daily bolus insulin, mean (IU/day)	35.6	15.6	35.6
Total daily basal insulin, mean (IU/day)	25.5	14.1	25.5
Total daily insulin—IU/kg/day	0.7 ± 0.1	0.42± 0.2	0.90 ± 0.2
Insulin sensitivity factor (IU/mg/dL)	1:29	1:68	1:29
Sensor Use (%)	98	96	85

## Data Availability

All data are available within this manuscript.

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
