# Peer review of "Smart Insulin Pen in Pregnant Women with Type 1 Diabetes: An Encouraging Case Series"

_healthcare, 2024, doi:10.3390/healthcare13010038_

Round 1
Reviewer 1 Report
Comments and Suggestions for Authors
The study is a work that depicts the importance of the use of technologically advanced medical devices especially in pregnancy. This is promising work for the individualization of MDI insulin therapy and monitoring, however, there are some points to ponder:
1. The abstract is sufficiently precise, but other aspects should also be covered like the result and conclusion.
2. The introduction is sufficient and well-referenced, however, the references were a bit less, try to search the literature for some more references not older than 5 years from the year of publication. This would show the impact of the study in the current health system.
3. The research Design could be improved by explaining the whole study design, why it was retrospective, and how the patients were using the devices.
4. The patients' convenience of the device use.
5. The number of subjects.? in line 41, "4" women were mentioned, but the cases presented were 3.
6. All the abbreviations in the manuscript should be at least fully mentioned once, to make the write-up readable.
7. Please correct the spelling of "especially" in line 190.
8. It would have been better if the cases were discussed in a simple way, for a better readers' understanding of the health issue. Like the importance and significance of the values of TIR and TAR and others and correlating them with the performance of the system under study.
Reviewer 2 Report
Comments and Suggestions for Authors
As attached

Reviewer 3 Report
Comments and Suggestions for Authors
In this interesting case series, Resi et al. described the use of smart insulin pens in three pregnant women. Despite the limited number of cases reported, the scarcity of data in the literature on this topic makes the manuscript of interest.
Some suggestions to improve the overall quality of the paper:
- Three clinical cases are described in the text, but the “Research Design and Methods” section refers to four women. This is somewhat confusing.
- Figures 1, 2, and 3: please pay attention to the language used in the captions at the bottom (it seems they were left in Italian).
- In Figures 1, 2, and 3, data on different TBR ranges (between 54 and 63 mg/dL and <54 mg/dL) are missing.
- For the three cases, I wonder what glycemic target was set in the system and what insulin sensitivity factor was chosen. This information seems critical for all three cases.
- The discussion appears to lack sufficient data (both RCT and real-world evidence) on the use of smart insulin pens in other populations (pediatric and non-pregnant adults).
- I suggest focusing the discussion on the patient education process when initiating smart insulin pens. It is particularly interesting that two out of the three cases presented did not use CHO counting.
- In my opinion, given this is a case series, the conclusions should be slightly mitigated
Comments on the Quality of English LanguageI recommend a substantial revision of the English language throughout the manuscript.
Round 2
Reviewer 3 Report
Comments and Suggestions for Authors
The authors have addressed all the reviewer’s comments. The quality of the manuscript has now significantly improved.